# Protective Effects of Melatonin against Carcinogen-Induced Oxidative Damage in the Thyroid

**DOI:** 10.3390/cancers16091646

**Published:** 2024-04-25

**Authors:** Jan Stępniak, Małgorzata Karbownik-Lewińska

**Affiliations:** 1Department of Endocrinology and Metabolic Diseases, Medical University of Lodz, Rzgowska St. 281/289, 93-338 Lodz, Poland; jan.stepniak@umed.lodz.pl; 2Polish Mother’s Memorial Hospital-Research Institute, Rzgowska St. 281/289, 93-338 Lodz, Poland

**Keywords:** thyroid, melatonin, oxidative stress

## Abstract

**Simple Summary:**

Melatonin is a hormone primarily produced in the pineal gland in response to darkness. Apart from regulating our body’s internal clock, it possesses strong antioxidant properties. Thanks to these characteristics, melatonin shows promise in preventing and treating various health issues, such as cancer. This review explores how melatonin can protect the thyroid gland from oxidative damage and, consequently, prevent the development of thyroid disease caused by documented or potential carcinogens. The article emphasizes the importance of melatonin in protecting the thyroid and how it can help to treat thyroid diseases, including cancer.

**Abstract:**

Melatonin, primarily synthesized in the pineal gland, plays a crucial role in regulating circadian rhythms and possesses significant antioxidative properties. By neutralizing free radicals and reducing oxidative stress, melatonin emerges as a promising agent for the prevention and therapy of many different disorders, including cancer. This paper reviews the relationship between the thyroid gland and melatonin, presenting experimental evidence on the protective effects of this indoleamine against oxidative damage to macromolecules in thyroid tissue caused by documented carcinogens (as classified by the International Agency for Research on Cancer, IARC) or caused by potential carcinogens. Furthermore, the possible influence on cancer therapy in humans and the overall well-being of cancer patients are discussed. The article highlights melatonin’s essential role in maintaining thyroid health and its contribution to management strategies in patients with thyroid cancer and other thyroid diseases.

## 1. The Thyroid Gland as an Organ of Oxidative Nature

The thyroid gland is a crucial endocrine organ responsible for controlling fundamental physiological processes such as metabolism, tissue/organ growth, and development in all vertebrates [1]. It contains two types of hormone-producing cells: (1) the major type, called thyroid follicular cells (or thyrocytes), which produce mainly thyroxine (T4) and, to a lesser extent, triiodothyronine (T3), and (2) the other type, called parafollicular cells (or C cells), in which calcitonin is synthesized.

The primary secretory product of thyrocytes, T4, is also called a prohormone; it requires conversion to the active T3 in peripheral tissues. T3 exerts effects on virtually all human cells possessing a nucleus and plays a crucial role in increasing the overall metabolic rate. Both T3 and T4 are iodothyronines, i.e., products of iodine incorporation into tyrosine residues within thyroglobulin (Tg).

Although the process of thyroid hormone synthesis is crucial for the entire organism, it can pose risks to the thyroid gland itself under special circumstances. This is because it involves a complex and multistep series of redox reactions utilizing hydrogen peroxide (H_2_O_2_), as the primary oxidizing agent, which is the basic form of reactive oxygen species (ROS). H_2_O_2_ acts as an oxidative equivalent in the following reactions catalyzed by thyroid peroxidase (thyroperoxidase; TPO): iodide oxidation, iodination of tyrosyl residues in Tg, and the subsequent phenoxy ether bond formation between pairs of iodotyrosines to generate iodothyronines. Because H_2_O_2_ is indispensable in the process of thyroid hormone formation and, therefore, should be available in unlimited quantities, it is probably present in the thyroid in amounts exceeding those required for iodine incorporation into thyroid hormones [2]. The fact that normal thyroid function depends on ROS and, therefore, this organ is constantly exposed to them, places the thyroid gland at risk of high oxidative stress. This phenomenon may justify the statement that the thyroid gland is an organ of “oxidative nature” [3] [Figure 1].

H_2_O_2_ is produced in the thyroid gland by NADPH oxidase/dual oxidase (NOX/DUOX), which are family transmembrane proteins. The thyroidal NOX/DUOX group comprises three enzymes: i.e., dual oxidase 1 (DUOX1), dual oxidase 2 (DUOX2) and NADPH oxidase 4 (NOX4). Of importance, DUOX1 and DUOX2 are typically active only at the apical plasma membrane of thyroid follicular cells [4]. Whereas DUOX2 is the main NOX/DUOX isoform catalyzing H_2_O_2_ formation for thyroid hormone synthesis [5,6], DUOX1 is probably active only under certain conditions associated with DUOX2 unavailability; however, its role in thyroid physiology is still not clear [7].

Contrary to DUOX enzymes, NOX4 is distinguished by its intracellular localization and, notably, by its continuous uncontrolled H_2_O_2_ production, which results solely from the gene expression of this enzyme [8]. The physiological function of H_2_O_2_ produced by NOX4 in the normal thyroid gland is not fully known. However, it is implied that NOX4 plays an important role in the regulation of redox processes occurring close to the basal cellular membrane. Studies indicate that H_2_O_2_ produced by NOX4 is potentially involved in the redox-sensitive control of thyroid differentiation [9,10]. It has been shown that TSH represses and iodine excess increases NOX4 mRNA expression [10]. NOX4-derived H_2_O_2_ also mediates endoplasmic reticulum signaling [11]. Therefore, taking into account the large amounts of secretory proteins required for thyroid functioning synthesized by the endoplasmic reticulum, the function of NOX4 may be of crucial importance [12].

Uncontrolled H_2_O_2_ production by NOX4 gains special significance when we consider that H_2_O_2_, in addition to serving as a substrate in hormone synthesis and signal transduction, can become a significant source of free radicals and other ROS. Because it is an oxidizing agent, H_2_O_2_ can induce damage to biological macromolecules such as DNA, lipids, and proteins. Additionally, H_2_O_2_ is a compound with a relatively long half-life (milliseconds to seconds) [13] and as a non-polar molecule has the capacity to diffuse through biological membranes, creating a hazard in a place far from its origin. Although H_2_O_2_ itself has limited reactivity towards macromolecules, it can give rise to highly reactive and most harmful free radical, i.e., hydroxyl radical (•OH), in a process that is catalyzed by transition metal ions, typically ferrous ion (Fe^2+^), which is known as the Fenton reaction (Fe^2+^ + H_2_O_2_ → Fe^3+^ + •OH + OH^−^). This reaction is particularly important in the thyroid gland, where heme iron is a component of TPO, a typical heme-containing enzyme. Therefore, iron is necessary for TPO biological activity and, consequently, for thyroid hormone synthesis.

Both experimental and epidemiological studies have shown that iron deficiency can negatively impact thyroid function [14]. It is known that thyroid hormone concentrations are lower in patients with iron deficiency, especially in certain patient groups such as pregnant women [15]. While the level of iron in the thyroid gland is tightly regulated, the fact that activated TPO is located at the apical membrane, exposing its heme-linked catalytic site to the thyroid follicular lumen, may result in an excessive amount of iron, which could potentially increase the risk of adverse oxidative reactions.

H_2_O_2_, through various mechanisms, may contribute to the formation of thyroid cancer [16]. Importantly, H_2_O_2_ may be involved in the co-occurrence of thyroid cancer and congenital hypothyroidism [17].

It is worth mentioning that in numerous experimental studies, Fenton reaction substrates, i.e., Fe^2+^ and H_2_O_2_, are repeatedly confirmed to increase oxidative damage to macromolecules in different tissues [18,19,20], such as the thyroid gland [21,22,23].

Given that the substrates of Fenton reaction are essential for thyroid hormone synthesis, yet—when in excess—can also inflict notable cellular harm and disrupt regular function, it is likely that thyroid epithelial cells possess a robust defense mechanism to counteract potential damage induced by free radicals. These protective mechanisms comprise antioxidative enzymes, such as superoxide dismutase (SOD), glutathione peroxidase (GPx), glutathione reductase (GR), catalase (CAT), and recently discovered peroxiredoxins [24]. In agreement with such an assumption, we made the following observations. The basal level of lipid peroxidation (LPO)—the index of oxidative damage to membrane lipids—was not higher in the thyroid gland than in other tissues [25]. Next, Fenton reaction-induced lipid peroxidation in the thyroid was lower than that in non-endocrine tissues such as the liver, kidney, brain cortex, spleen, small intestine [20], and ovary [25]. However, under conditions associated with endogenous abnormalities or with exposure to exogenous pro-oxidative agents, antioxidative machinery may become inefficient, resulting in huge oxidative damage and, consequently, in the development of various diseases, including cancer [26]. Interestingly, our studies also indicate that under basal conditions, female thyroid cells are exposed to higher concentrations of H_2_O_2_, most likely due to the activity of NADPH oxidases, primarily NOX4, and to a lesser extent, DUOX1 and DUOX2 [27], which can explain—at least partially—the higher prevalence of thyroid diseases, such as cancer, in the female population.

It should also be stressed that iodine, the fundamental building component of thyroid hormones, serves as a potent oxidizing agent. It can act as both an antioxidant and a prooxidant, depending on its chemical form. On the one hand, iodide ion (I^−^) is a potent antioxidant, the reducing properties of which make it an important free radical scavenger [28]. On the other hand, hypoiodite anion (IO^−^), which is oxidized iodine, is a potent oxidant with strong bactericidal activity. Moreover, an excess of iodine can interfere with thyroid hormone synthesis and, therefore, can act as a potential endocrine-disrupting chemical (EDC, endocrine disruptor), and this action occurs—at least partially—via mechanisms involving oxidative stress (reviewed in detail in [29]).

## 2. Melatonin as an Antioxidant—The Short Overview

Melatonin (N-acetyl-5-methoxytryptamine) is a ubiquitous molecule with a broad presence in nature, synthesized by a multitude of living organisms. It is postulated to have emerged on Earth approximately 3.2 to 3.5 billion years ago in photosynthetic cyanobacteria. Its synthesis in these organisms served as an essential antioxidant mechanism to counteract the production of harmful ROS during the photosynthetic process [30,31]. In accordance with the endosymbiotic theory regarding the origin of eukaryotic organelles [32], these organisms became integral components of eukaryotic physiology, together with the melatonin synthesis machinery [33]. These attributes, having conferred notable advantages to the functionality of the emerging eukaryote, were subsequently conserved and, throughout the course of evolution, were adapted for a number of other critical functions [31]. Nevertheless, the original antioxidative function of melatonin was retained and persists in present-day mammals, including humans.

In mammals, the major site for melatonin synthesis is the pineal gland. In addition, melatonin is produced to a lesser extent by numerous peripheral organs, including the bone marrow, lymphocytes, eyes, gastrointestinal tract, and human and rodent skin. Melatonin synthesis and secretion are suppressed by light and enhanced by dark [34]. The synthesis of melatonin involves a multi-step process, commencing with tryptophan as the precursor, followed by successive enzymatic conversions leading to the formation of serotonin, and culminating in the conversion to melatonin [35]. It should be stressed that melatonin is a lipophilic compound that can easily penetrate biological membranes [36].

Melatonin achieves its antioxidative action via a variety of means. Melatonin effectively neutralizes a broad spectrum of free radicals, including the highly reactive and destructive •OH, alkoxy radicals (RO•), peroxy radicals (ROO•), and nitric oxide (•NO) [37]. In addition, melatonin also reportedly chelates transition metals, which are involved in the Fenton reaction; in doing so, melatonin reduces the formation of •OH [38]. Additionally, it targets various non-radical oxidants like H_2_O_2_, singlet oxygen (^1^O_2_), and peroxynitrite anion (ONOO−) [39]. As an electron-rich molecule, melatonin directly interacts with ROS, subsequently producing the following stable metabolites: cyclic 3-hydroxymelatonin, N1-acetyl-N2-formyl-5-methoxykynuramine, and N1-acetyl-5-methoxykynuramine [36,37]. Each of these metabolites also functions as a free radical scavenger. As a result, a single melatonin molecule along with its metabolites can neutralize numerous ROS [36,37]. Moreover, melatonin and its metabolites, contrary to antioxidants such as vitamin C, vitamin E, and glutathione (GSH), do not undergo redox cycling, and, thus, they do not promote secondary oxidation [36]. Also of importance, end-products of antioxidative action of melatonin are stable and excreted in the urine [40].

Besides directly neutralizing free radicals, melatonin can stimulate antioxidative enzymes such CAT, SOD, GPx, and GR [41,42,43]. At the same time, melatonin suppresses the activity of pro-oxidant enzymes, such as inducible nitric oxide synthase [44] or xanthine oxidase, the latter enzyme generating ROS produced after exposure to microwave radiation [45].

The physiological blood concentration of melatonin in humans depends on several factors, especially light and dark exposure. During the daytime, it is found to be 0–20 pg/mL, and at night it can reach levels as high as 40–200 pg/mL [46]. The nocturnal melatonin peak generally decreases with age, although there are notable individual variations. Some elderly individuals exhibit rather low nighttime melatonin levels that are nearly identical to their daytime levels, while others maintain a distinct melatonin rhythm with only modest reductions in nocturnal values [47]. The melatonin level in organisms can also be influenced by the consumption of melatonin-rich foods [48] or administration of aminoacids required for melatonin production [49]. Additionally, melatonin is increasingly being used as an over-the-counter dietary supplement mainly for the treatment of circadian rhythm sleep disorders [50]. Furthermore, it has gained recognition as a supplement that supports conventional drug therapies in various conditions, such as multiple sclerosis [51] and cancer [52]. Of importance, melatonin has a very good safety profile, which was also documented when this indoleamine was applied at very high doses. Some mild side effects, such as sleepiness, nausea, dizziness, and headache, have been observed exclusively during long-term treatment [53]. Usually recommended daily doses of melatonin range from 2 to 10 mg. In turn, as a high dose as 25 mg was used in clinical trials [54]. It is worth mentioning that intravenous administration of melatonin at a dose of 25 mg resulted in its blood concentration of approximately 7.52 × 10^5^ pg/mL [54].

It is worth stressing that melatonin stands out among various hormones implicated in thyroid health due to its potent antioxidative effects, which are critical in the context of thyroid disorders. Unlike many other hormones that influence thyroid function through direct regulatory mechanisms, melatonin primarily contributes through its ability to modulate oxidative stress, a key factor in thyroid physiology and probably thyroid pathology.

Thyroid disorders, including both hypothyroidism and hyperthyroidism, are often accompanied by an increased oxidative stress burden due to abnormal metabolic rates associated with synthesis, degradation, and peripheral effects of thyroid hormones. The fact that melatonin is a highly effective free radical scavenger and is able to stimulate antioxidative enzymes but inhibit prooxidative enzymes means this indoleamine can be particularly beneficial in the context of thyroid diseases. It helps to protect thyroid cells from oxidative damage, which can lead to cellular dysfunction or death, and plays a role in preventing the onset and progression of thyroid diseases. Therefore, it is highly probable that melatonin affects the thyroid gland using mostly mechanisms different from endocrine mechanisms, which distinguishes this hormone from other hormones acting via binding to specific receptors. Summarizing this issue, the potent antioxidative properties of melatonin set it apart from other hormones that influence thyroid function, offering a distinctive therapeutic potential in managing thyroid disorders.

Antioxidative actions of melatonin are presented in Figure 2.

## 3. Relationship between the Thyroid and Melatonin

The interplay between melatonin, a hormone governing circadian rhythms and sleep-wake cycles, and the thyroid gland, a key component of the endocrine system regulating metabolism, reveals intricate physiological connections. Earlier studies clearly showed the profound impact of melatonin on the thyroid gland, indicating its direct inhibitory effects on both growth processes and thyroid hormone synthesis (reviewed in detail in [55]). It was also observed that melatonin is able to affect thyroid function and thyroid hormone concentrations in the blood, acting at the level of the hypothalamus, the pituitary gland, and/or the thyroid gland [56]. In photoperiodic species, namely animals that rely on day length or night length for biological and behavioral changes such as growth, migration, reproduction, and rest periods, melatonin coordinates the local control of thyroid hormone metabolism within the medio-basal hypothalamus, thereby initiating the photoperiodic response [57,58,59,60].

Melatonin can have also a direct effect on the activity of the thyroid gland. Interestingly, C cells but also thyroid follicular cells express mRNAs of the key enzymes in melatonin biosynthesis (i.e., aralkylamine N-acetyltransferase (AANAT) and acetyl serotonin methyltransferase (ASMT)), and they also possess melatonin receptor (MT_1_) [61]. It has also been observed that thyroid C cells synthesize melatonin under TSH control [62]. The significance of both phenomena remains to be elucidated.

Further investigations have found that daily administration of melatonin decreased T3 and T4 concentrations in dogs [63] and also suppressed thyroid hormone secretion in hyperthyroid rats [64]. Conversely, in the study on rats with propylthiouracil-induced hypothyroidism, a significant decrease in melatonin levels has been observed [65]. These findings underscore a complex and bidirectional interaction between melatonin and thyroid hormone homeostasis.

The relationship between the thyroid and melatonin documented in experimental studies is presented in Figure 3.

## 4. Potential Protective Effects of Melatonin against Carcinogenesis in Humans, Human Tissues, or Human-like Cell Lines

Due to such properties of melatonin as antioxidative, anti-inflammatory, immune-modulating, and endocrine effects, this indoleamine may also modulate the process of carcinogenesis. Indeed, it was found to exhibit anticarcinogenic action via antiproliferative, antiangiogenic, apoptotic, and metastasis-inhibitory pathways in experimental models. For example, antiproliferative effects of melatonin occurred via downregulation of oncogenic factors, such as cyclin D1 in MCF-7 breast cancer cells [66] or of histone deacetylase 9 (HDAC9) in non–small cell lung cancer [67]. The antiangiogenic properties of melatonin were demonstrated in human HepG2 liver cancer cells, where this indoleamine interferes with vascular endothelial growth factor (VEGF) [68]. Its ability to inhibit metastasis is evident in breast and lung cancers, where it disrupts key signaling pathways such as DJ-1/KLF17/ID-1 and epithelial-mesenchymal transition, respectively; the latter is mediated through the MT1 receptor and several downstream effectors including PLC, p38/ERK, and β-catenin [69,70]. In turn, in studies in humans with solid or liquid tumors, melatonin improved the sensitivity to conventional treatment [71,72].

Although there is a considerable amount of experimental research on the anticancer therapeutic properties of melatonin, studies involving humans are comparatively limited. Moreover, considering the wide spectrum of cancer types, anticancer therapies, and the range of melatonin actions, reaching a consensus about potential advantageous effects of melatonin applied in humans is challenging. The following examples of the therapeutic effects of melatonin in cancer patients are worth mentioning: a higher rate of tumor regression in patients with colorectal and gastric cancer [73], a higher survival rate in patients with solid neoplasms [74], and improvement of the chemotherapeutic effects of 5-fluorouracil [73], frequently employed in the treatment of various neoplasms. Another example of using melatonin as an anticancer adjuvant is its co-administration with platinum-based neoadjuvant chemotherapy for oral squamous cell carcinoma. This study demonstrated that administering 20 mg/daily of melatonin for seven days before and during therapy could reduce the expression of hypoxia-inducible factor (HIF-1α), which is a transcription factor contributing to cancer development during hypoxic conditions [75]. Melatonin co-administrated with chemotherapy has also reduced residual tumor percentage; however, it did not affect clinical response to neoadjuvant chemotherapy [75].

Another equally important aspect of melatonin’s influence on cancer therapy is its potential to have a broad impact on the adverse effects of anticancer medications and on the overall well-being of cancer patients. An example is the neuroprotective effects of melatonin counteracting adverse effects of adjuvant chemotherapy applied in breast cancer patients [76].

In our studies, we have observed that melatonin can completely prevent oxidative damage to membrane lipids caused by sorafenib or lenvatinib (multi-targeted tyrosine kinase inhibitors approved to treat advanced hepatocellular carcinoma, renal cell carcinoma, and radioiodine-refractory differentiated thyroid carcinoma) in noncancerous porcine tissues of the thyroid, liver, and kidney [77]. As both drugs can cause adverse effects resulting partially from ROS generation [78], melatonin can be considered to be applied in co-treatment with these tyrosine kinase inhibitors to prevent their undesirable toxic effects occurring via oxidative stress.

Although there has not been a dedicated study examining melatonin as a treatment for thyroid cancer in humans, melatonin shows some therapeutic effects in experimental studies.

As was mentioned above, melatonin minimizes thyroid cell damage due to irradiation [79,80]. On the other hand, some recent studies have proposed that melatonin can enhance the therapeutic effects of radiotherapy by acting as a radiosensitizer in thyroid cancer cells [81].

In another study with the use of a cell line from anaplastic thyroid cancer, melatonin exerted a dose-dependent cytotoxic effect. It significantly decreased cell viability and induced cell reproductive death at concentrations greater than 1 mM [82]. Such findings suggest that melatonin could be valuable as an adjuvant in anaplastic thyroid cancer therapy.

Research on a mouse model of thyroid cancer has demonstrated that melatonin treatment increased the production of the hydrogen sulfide (H_2_S) signaling molecule, which in turn increased the expression of pro-apoptotic tumor suppressor, resulting in the activation of cell apoptosis and suppression of cancer cell proliferation [83].

It is worth considering to what extent melatonin would be effective in particular types of thyroid cancer. The mechanisms behind the anticancer effects of melatonin are not sufficiently understood to allow for comparison across such diverse cancers as papillary and follicular cancer, both differentiated thyroid cancer formed from thyroid follicular cells and medullary thyroid cancer formed from C cells. Additionally, both types of thyroid cells possess the machinery for thyroid hormone synthesis as well as melatonin receptors. Therefore, it is currently challenging to conclusively determine if the effects of melatonin may vary specifically among these types of thyroid cancer due to the limited understanding of its underlying anticancer mechanisms.

The mechanisms of the potential protective effects of melatonin against the process of carcinogenesis are presented in Figure 4.

## 5. Evidence on Protective Effects of Melatonin against Oxidative Damage to Macromolecules Caused by Documented/Potential Carcinogens in the Thyroid

### 5.1. Melatonin and Oxidative Stress in the Thyroid

Thus summarizing results published so far on the protective effects of melatonin against oxidative damage caused by documented or potential carcinogens is justified. Whereas it is reasonable to consider such an action of melatonin in different tissues, the thyroid gland does constitute such an organ, which is characterized by relatively high levels of oxidative stress. Various in vitro and in vivo models were used to study the protective effects of melatonin against unfavorable oxidative processes in the thyroid. Regarding studies in humans, only a few publications on this issue are available in the literature.

Different aspects of oxidative stress were examined, including oxidative damage to macromolecules (such as membrane lipids, proteins, DNA, and other nucleic acids), activities of pro- and antioxidative enzymes, and levels of free radicals and other reactive oxygen species.

Taking into account the “oxidative nature” of the thyroid and the strong antioxidative properties of melatonin, it is especially interesting to investigate the extent to which melatonin affects favorably oxidative damage in this organ. Foremost, several in vitro studies performed by our team have shown that melatonin alone did not reduce LPO levels below the physiological threshold in the thyroid gland [20,21,25,84,85]. No changes in LPO levels in response to melatonin exposure were also observed in other tissues such as the spleen, liver, brain cortex, kidney, or intestine both in vitro and in vivo [20,86,87]. This action may offer an additional advantage since oxidative processes occur with a certain intensity in biological structures and are essential for physiological processes. However, these results neither imply that melatonin is ineffective as an antioxidant nor exclude its potential to prevent oxidative changes resulting from the action of any pro-oxidant. In fact, in the case of oxidative damage caused by pro-oxidative agents, we have observed that melatonin is an effective protective antioxidant in the thyroid gland.

In most studies cited below, melatonin revealed protective effects when used in concentrations ranging from nM to mM under in vitro conditions or when used in doses from 1 to 10 mg/kg under in vivo conditions.

### 5.2. Protective Effects of Melatonin against Oxidative Damage Caused by Documented Carcinogens (Listed in the International Agency for Research on Cancer (IARC) Monographs)

#### 5.2.1. Hexavalent Chromium/Potassium Dichromate

Hexavalent chromium (Cr(VI)) is a highly toxic form of chromium used in various industrial processes. Exposure to Cr(VI) is prevalent in industries such as stainless steel production, chrome plating, leather tanning, and chemical dye production. Exposure primarily occurs through inhalation of contaminated air or ingestion of water polluted with Cr(VI).

Chromium (VI) compounds are classified by the International Agency for Research on Cancer (IARC) as Group 1 carcinogens [88], indicating that they are carcinogenic to humans. The following cancer types are best documented to be associated with the exposure to Cr(VI): cancers of the lung, nasal cavity, and paranasal sinuses; however, Cr(VI) may also increase the risk of brain cancer, malignant lymphoma, and tumors of the endocrine glands [89]. Cr(VI) induces cellular damage and adverse health effects through various mechanisms such as oxidative stress, DNA damage, dysregulating gene expression, and cancer-related signal transduction [90]. Upon entry into cells, Cr(VI) is reduced by ascorbate and biological thiols such as GSH; this process generates H_2_O_2_ and other free radical species that may contribute to increased oxidative stress causing damage to cellular lipids, proteins, and DNA [91]. Exposure levels to Cr(VI) can range from background environmental levels of less than 1 microgram per cubic meter (µg/m^3^) to over 100 µg/m^3^ in occupational settings involving chrome plating, stainless steel welding, and manufacturing processes.

As demonstrated in studies on adult male albino rats, the administration of potassium dichromate (K_2_Cr_2_O_7_) (a hexavalent chromium compound), Cr(VI)) in the dose of 25 mg/kg/day for 2 months leads, among other effects, to follicular hyperplasia and follicular enlargement in the thyroid [92]. In this study, the simultaneous administration of melatonin in the dose of 10 mg/kg/day demonstrated a strong protective effect, as thyroid follicular cells from animals treated with K_2_Cr_2_O_7_ plus melatonin appeared ultrastructurally similar to those of the control group. The above effect surpassed that of vitamin C, highlighting melatonin’s potential as a mitigating factor against the adverse impacts of Cr(VI) exposure on thyroid function [92].

#### 5.2.2. Radiotherapy/Ionizing Radiation

Radiotherapy is a type of cancer treatment that uses high-energy electromagnetic radiation (such as gamma or X-rays) or radiation of particles (such as protons, neutrons, and high-speed electrons). Passing through the cells, this ionizing radiation can cause the displacement of electrons from atoms, which, in turn, produces biological effects by breaking chemical bonds and forming highly reactive free radicals. This process can induce DNA damage either by direct means (DNA damage directly caused by ionization) or indirectly (through the production of free radicals), consequently leading to cell death.

According to the IARC classification, all types of ionizing radiation belong to Group 1 carcinogens (i.e., carcinogenic to humans) [93]. Since radiotherapy is commonly employed in patients with head and neck cancer or breast cancers, and also lymphomas and central nervous system tumors, the radiation in these regions often affects the thyroid gland. It is worth mentioning that the most common complication that can occur after radiation therapy for breast or head and neck cancers is hypothyroidism [94], but in rare cases, new systemic and local cancers, including thyroid cancer, can appear [95]. Radiation therapy doses, measured in Gray (Gy), for treating various cancers range from 10 to 70 Gy, depending on cancer type, stage, treatment goal, and area treated.

Studies conducted on rats have shown that radiation of the neck region with a total dose of 16–18 Gy led to increased inflammation, vacuolization, degradation, swelling, and necrosis in the thyroid gland [79]. Additionally, there were elevated levels of tumor necrosis factor-α (TNF-α), interleukin-1β (IL-1β), thiobarbituric acid reactive substances (TBARS), and nitric oxide (NO) [80]. Melatonin at doses of 10–50 mg/kg administered 10–15 min before exposure to radiation caused a significant reduction of these unfavorable changes in histopathological and biochemical parameters [79,80]. Moreover, GSH values were higher in the melatonin plus radiotherapy group compared to radiotherapy alone [80].

It is worth mentioning that the antioxidative effects of melatonin in protection against cellular damage caused by ionizing radiation were summarized over 20 years ago by one of the authors of the present review [96]; however, the thyroid tissue had not been examined in these earlier studies.

#### 5.2.3. Radioiodinetherapy/Iodine-131

Radioisotopes of iodine are called radioiodine or radioactive iodine (RAI). Iodine-131 (I-131) is a radioisotope of iodine that primarily emits β rays and—to a much lesser extent—γ rays. With a relatively short half-life of just over 8 days, it remains the radionuclide of choice for both therapeutic and diagnostic applications. I-131 therapy is indicated for the management of hyperthyroidism and thyroid cancer. Its action is dependent on the uptake of iodine by thyroid tissue. Once absorbed by thyroid follicular cells, I-131 emits β rays, a form of ionizing radiation that induces local damage to thyroid follicular cells.

Radioiodines, including I-131, have been classified by the IARC as Group 1 carcinogens, indicating their carcinogenicity to humans [97]. In the case of RAI therapy for hyperthyroidism, the most recent meta-analysis revealed no significant elevation in risk for specific types of cancer, with the exception of thyroid cancer incidence, for which the risk was statistically higher [98]. Regarding RAI therapy for differentiated thyroid cancer, it has been found that this treatment in childhood and young adulthood was associated with an increased risk of solid cancer (by 23%) and leukemia (by 92%), particularly more than 20 years after exposure [99].

Due to these undesirable effects of radioiodine, it is important to determine possible protective factors. In animal studies conducted on rats, melatonin in a dose of 12 mg/kg/day administered 2 days before and for 1 week after oral RAI prevented adverse effects of a single dose of 111 MBq (3 mCi) RAI on spermatozoa quality [100] and on the oxidative stress parameters in the liver [101]. A capsule containing 300 mg of melatonin powder given 1 h before the administration of iodine-131 (370 to 740 MBq (10 to 20 mCi)) slightly reduced (without statistical significance) the rate of I-131-induced chromosomal damage in lymphocytes from hyperthyroid patients [102].

Regarding isotope I-123, which emits mostly γ radiation and is used almost exclusively for diagnostic imaging, no results are found in the literature on its damaging effects on macromolecules.

#### 5.2.4. Night-Shift Work

Night-shift work is defined as work occurring during the regular sleeping hours of the general population. According to the IARC classification, night-shift work belongs to Group 2A (probably carcinogenic to humans) [103]. There is sufficient evidence in experimental animals for the carcinogenic effects of alterations in the light–dark schedule. However, evidence of the carcinogenic effects of night-shift work in humans is limited. The hypothesized contributing factors are directly related to disruptions in the circadian rhythm and decreased melatonin production, both considered oncogenic in nature [104]. The circadian rhythm is a fundamental internal biological process, which regulates crucial mechanisms such as metabolism, DNA repair, and immune system function, all integral to cancer pathogenesis. The circadian rhythm enables human bodies to function optimally in a 24-h cycle.

The IARC Working Group based its decision to classify night-shift work as probably carcinogenic to humans (Group 2A) on evidence primarily derived from studies on breast cancer risk and, to a lesser extent, on prostate and colorectal cancer risk [103]. Studies examining the correlation between night-shift work and other types of cancer are limited. Regarding the correlation between night-shift work and thyroid cancer risk, a study among nurses found no direct link [105]. However, the research did suggest a potential association between persistent sleep difficulties among night-shift workers and a higher risk of thyroid cancer. Notably, nurses with over a decade of night-shift experience, coupled with frequent sleep difficulties, showed a modestly increased risk of thyroid cancer [105]. Additionally, in a large US cohort study, it was found that exposure to light at night (LAN) was positively associated with thyroid cancer risk [106]; this relationship may be directly related to the fact that exposure to artificial LAN inhibits nighttime secretion of melatonin and subsequently may cause circadian disruption [107].

Despite the well-established role of melatonin in circadian rhythm regulation, there has not been a dedicated study examining melatonin as a preventive treatment for night-shift workers. This underscores the need for further research to explore the potential protective effects of melatonin in mitigating the adverse health impacts associated with night-shift work, particularly in the context of cancer risk.

#### 5.2.5. Nitrobenzene

Nitrobenzene is a synthetic chemical not found naturally in the environment. It is widely used in industry, primarily in the synthesis of aniline—a commodity chemical produced on a very large scale and used as a precursor of polyurethane, dyes, and other industrial chemicals. Given its extensive industrial usage, nitrobenzene is ubiquitously present in the environment, increasing the likelihood of hazardous exposure.

Nitrobenzene is classified by the IARC as a Group 2B compound (possibly carcinogenic to humans) [108]. It has been demonstrated that chronic inhalation exposure of experimental animals to nitrobenzene is associated with an increased incidence of tumors at multiple sites, including follicular thyroid tumors [109]. The U.S. Environmental Protection Agency, based on studies in rats and mice, determined that the Reference Dose (RfD) for nitrobenzene is 0.0005 mg/kg/day [110]. This RfD represents an estimate of a daily human exposure (including sensitive subpopulations) that is likely to be without a significant risk of adverse effects over a lifetime. The observed toxic effects of nitrobenzene include an increased incidence of thyroid follicular adenomas in rats and mice. Additionally, endometrial tumors were reported in the same species [111].

In the research conducted by our team, nitrobenzene in concentrations of 7.5 and 10.0 mM increased LPO levels in porcine thyroid homogenates [112]. In this study, we have shown that melatonin in concentrations as low as 0.0001 mM effectively prevented LPO damage caused by nitrobenzene (7.5 mM) [112]. It should be stressed that such low concentrations of melatonin (0.0001 mM) are only two orders of magnitude higher than physiological melatonin blood concentration in humans i.e., 0.0001 mM vs. 0.000001 mM [46].

#### 5.2.6. Bromium/Potassium Bromate

Potassium bromate (KBrO_3_) has been classified by the IARC as a compound belonging to Group 2B carcinogens (a possible human carcinogen) [113]. KBrO_3_ is a strong oxidizing agent. It was commonly used in the past as a food additive in the bakery industry, but in some countries industrial use of KBrO_3_ has persisted.

Oxidative properties of KBrO_3_ play a fundamental role in its carcinogenic action. This compound increases the formation of free radicals and reactive oxygen species (ROS), such as peroxynitrite anion (ONOO^−^) or nitric oxide (NO•), and—simultaneously—it decreases the activity of antioxidative enzymes, such as GPX [114]. KBrO_3_ was found to cause oxidative DNA damage, as measured by the increased level of 8-hydroxy-2′-deoxyguanosine (8-OHdG), and to induce mutations (GC to TA transversions) in the rat kidney [115].

In studies conducted by our team, we investigated the effects of melatonin and another indole substance possessing a chemical structure similar to that of melatonin, indole-3-propionic acid (IPA), on LPO induced by KBrO_3_. We found that neither melatonin (at concentrations ranging from 0.01 mM to 7.5 mM) nor IPA (at concentrations ranging from 0.01 mM to 10 mM) reduced LPO induced by 5 mM KBrO_3_ in porcine thyroid homogenates [116]. However, when melatonin or IPA was used in vivo in rats as a pretreatment, it decreased LPO induced by a single injection of KBrO_3_ in the dose of 110 mg/kg in the thyroid gland [116]. This protective effect was achieved by administering melatonin or IPA at a dose of 0.0645 mmol/kg (i.e., 15 mg/kg) b.w. twice daily for 10 days. It is worth mentioning that not only in vivo but also in vitro damaging effects of KBrO_3_ were prevented by propylthiouracil, a well-known antithyroid medication that possesses certain antioxidative properties [116]. However, taking into account clinical applications of the above results [116], only melatonin or IPA can be considered in preventive actions, as propylthiouracil would cause undesired side effects such as hypothyroidism.

#### 5.2.7. Mercury

Mercury (Hg) is one of the top ten chemicals of major public health concerns as stated by the World Health Organization (WHO) [117]. This chemical is a well-known toxic metal, causing occupational but also accidental exposures and, consequently, damage in human and animal organs. Although several observations link Hg exposure to cancer, the scientific evidence regarding the potential role of Hg in carcinogenesis is not clear [118].

According to the IARC classification, inorganic Hg species belong to Group 3 agents (agents with inadequate evidence for carcinogenicity in humans and experimental animals) [119]. In contrast, methylmercury (MeHg)—a compound formed from inorganic mercury by the action of microbes that live in aquatic systems—is categorized by the IARC as a Group 2B compound (potentially carcinogenic to humans) [119]. Although the RfD for elemental mercury was not assessed, it was determined to be 0.1 µg/kg/day for methylmercury [120].

Due to their toxic properties, mercury and its derivatives can induce oxidative damage, genotoxicity, and autoimmune reactions. Such adverse phenomena have also been observed in the thyroid gland [121,122]. For instance, it has been shown that occupational exposure to mercury is associated with RNA oxidative damage, as measured by the increased level of 8-oxo-7,8-dihydroguanosine (8-oxoGuo) [123]. These findings suggest a potential role of mercury and its derivatives in the pathogenesis of thyroid cancer, autoimmune thyroiditis, and hypothyroidism. A recently published meta-analysis has revealed an association between mercury exposure and thyroid cancer risk, implying a possible predisposing factor; however, further research is necessary to define the clinical relevance of this relationship [124].

In studies on adult male albino rats of the Wistar strain, it has been shown that the administration of mercury chloride (inorganic mercury compound) in the dose of 2–4 mg/kg b.w. caused a significant increase in oxidative stress in the thyroid gland imposed by a significant decline in levels of antioxidative enzymes such as SOD, CAT, GPX, GR, and non-enzymatic antioxidant GSH, followed by an elevated level of LPO [125]. Co-administration of melatonin in the dose of 5 mg/kg b.w. partially protected against mercury-induced changes [125].

Protective effects of melatonin against oxidative damage caused by documented carcinogens (listed in the International Agency for Research on Cancer (IARC) Monographs) are summarized in Table 1 and presented in Figure 5.

### 5.3. Protective Effects of Melatonin against Oxidative Damage Caused by Potential Carcinogens (Not Listed in the International Agency for Research on Cancer Monographs)

#### 5.3.1. Potassium Iodate

Iodine is an essential element for the synthesis of thyroid hormones. Iodine deficiency can lead to hypothyroidism and, consequently, to metabolic and developmental disorders, and it is associated with a higher risk of follicular thyroid cancer. Therefore, many countries have implemented iodine prophylaxis programs to eliminate iodine deficiency. Although excessive iodine intake may induce thyroid dysfunction and increase the relative risk of papillary thyroid cancer, the benefits outweigh the risks [126].

In most programs of iodine prophylaxis, iodized salt is used with either potassium iodide (KI) or potassium iodate (KIO_3_), as iodine carriers. It is known that these two iodine compounds possess distinct pro- and antioxidative properties. Unlike KI, KIO_3_ acts as an oxidant and can readily react with oxidizable substances [29]. 

Despite the chemical similarity of KIO_3_ to the previously described KBrO_3_, studies on the oxidative properties of iodate have shown that it has low, if any, genotoxic potential [127]. For this reason, KIO_3_ is not included in the IARC carcinogen list. However, in our study, we found that KIO_3_ can induce oxidative damage to membrane lipids in homogenates of several tissues, including thyroid tissue.

We have observed that KIO_3_ at concentrations of 20, 15, 10, 7.5, and 5 mM increased LPO in various porcine tissues, including the thyroid, ovary, liver, kidney, brain, spleen, and small intestine. Notably, the damaging effect of KIO_3_ at 10 and 7.5 mM was lower in the thyroid than in other tissues, and the lowest concentration of 5 mM did not exhibit any damaging effect in the thyroid. It is worth mentioning that these concentrations of KIO_3_ are close to the physiological iodine levels in the thyroid (approx. 9.0 mM). Whereas melatonin (5 mM) reduced LPO induced by 10, 7.5, and 5 mM of KIO_3_ in all tissues, in the thyroid it was additionally protective against as high a concentration of KIO_3_ as 15 mM, indicating that melatonin has a particularly beneficial role in the thyroid gland. Furthermore, LPO levels in the thyroid (vs. other tissues) were found to be lower when exposed to KIO_3_ in conjunction with melatonin. This disparity in LPO levels may suggest that the damaging effect of KIO_3_ is weaker in the thyroid than in other tissues, highlighting the tissue-specific protective role of melatonin against oxidative stress [25,84,85]. Moreover, cumulative protective effects of melatonin (5 mM) and IPA (10 mM) in the thyroid homogenates were stronger than those revealed by each of these two antioxidants used separately [84]. Interestingly, we have also observed that potassium iodide (KI), another iodine compound used in salt iodization programs, did not induce LPO in the thyroid when used in concentrations similar to those, in which KIO_3_ revealed damaging effects; moreover, it even reduced oxidative damage induced by the Fenton reaction substrates [128]. Our observations suggest that melatonin could be an effective agent in preventing oxidative damage and potentially reducing the risk of thyroid cancer formation caused by iodine compounds used in iodine prophylaxis.

It is worth mentioning that iodine when in excess is considered to be not only a potential carcinogen but also a potential endocrine disruptor [29].

#### 5.3.2. Iron-Induced Oxidative Damage

Iron, a crucial micronutrient, acts as a cofactor in fundamental cellular processes, including oxygen transport, cell proliferation, and energy metabolism. However, an imbalance leading to iron overload can induce heightened oxidative stress, potentially resulting in adverse effects. Although iron is not considered by the IARC as a human carcinogen, numerous animal models unequivocally indicate that an excess of iron can contribute to carcinogenesis. This is further substantiated by a range of human epidemiological data concerning cancer risk and prognosis [129].

Our first study on the potential protective effects of melatonin against experimentally induced oxidative damage in the thyroid was performed with the use of a classic experimental model. In the in vitro study, we applied Fenton reaction substrates, i.e., ferrous ions (Fe^2+^), in the concentration of 40 µM and hydrogen peroxide (H_2_O_2_) in the concentrations of 0.5 mM to induce LPO in porcine thyroid homogenates [21]. Expectedly, Fenton reaction substrates increased the level of LPO, and this damaging effect was prevented by melatonin in a concentration-dependent manner, with complete reduction to a control level when melatonin was added in the concentration of 5 mM. Interestingly, in our more recent work, we demonstrated that melatonin in the concentration of 5 mM is also highly effective in protecting against oxidative damage induced by extremely high concentrations of iron, i.e., as high as 4.8 mM. The protective effects of melatonin were observed not only in porcine thyroid but also in various non-endocrine porcine tissues (liver, kidney, brain cortex, spleen, and small intestine) [20]. The high iron concentrations used in our study [20] correspond to those found in patients with iron overload caused by congenital disturbances of iron metabolism (hemochromatosis) and by secondary hemochromatosis resulting from repeated blood transfusions or overconsumption of iron. Local iron overload can also be caused by chronic hepatitis C, ovarian endometriosis, or asbestos exposure [130]. This is particularly important when we take into account the fact that the carcinogenicity of iron has been clearly shown both in animal models and human studies [131]. In fact, hemochromatosis, chronic hepatitis C, or ovarian endometriosis are associated with an increased risk of cancer [130].

#### 5.3.3. Endocrine Disruptors

Endocrine disruptors are exogenous chemicals or mixtures of chemicals that interfere with any aspect of hormone action [132]. Compounds such as ammonium thiocyanate (NH_4_SCN), sodium fluoride (NaF), potassium selenocyanate (KSeCN), sodium chlorate (NaClO_3_), potassium perchlorate (KClO_4_), potassium nitrate (KNO_3_), and bisphenol A (BPA) pose significant risks to thyroid health. They can disrupt iodine uptake through inhibition of the sodium/iodide symporter (NIS) activity, which is crucial for active iodine transport into the thyroid; consequently, they inhibit thyroid hormone synthesis and metabolism [133]. They are shortly called NIS inhibitors. Although these endocrine disruptors are not specifically listed as carcinogens by the IARC, it has been observed that they can act through the induction of oxidative stress, and they are generally considered potential carcinogens. Therefore, in our recent study, we examined the damaging effects of these NIS inhibitors and the potential protective effects caused by melatonin and IPA [134].

Ammonium thiocyanate (NH_4_SCN) is utilized across a wide range of industries and chemical processes. It is commonly employed in the production of herbicides and thiourea and as a stabilizing agent in textile and dyeing processes. Its metabolic transformation in organisms may lead to the formation of reactive species, such as thiocyanate ions (SCN^−^), which can participate in reactions generating ROS [135]. In our in vitro study, NH_4_SCN used in concentrations of 250–500 mM increased LPO levels in thyroid homogenates, while both melatonin and IPA, at concentrations of 5 mM, completely prevented these changes [134].

Sodium fluoride (NaF) is a chemical compound recognized for its role in promoting dental health, being widely used in toothpaste and drinking water to prevent tooth decay by strengthening tooth enamel. It also serves various industrial purposes, including acting as a flux in the manufacture of aluminum and as an insecticide. The toxicity caused by fluoride is attributed to increased oxidative stress and altered antioxidant defense mechanisms [136,137]. In the in vitro study performed by our team, NaF in concentrations of 25–100 mM induced LPO in thyroid homogenates. As expected, both melatonin (5 mM) and IPA (5 mM) completely prevented NaF-induced oxidative damage to membrane lipids [134].

Selenium (Se) is an essential trace element that plays a vital role in various biological processes. Its main biological role is related to its incorporation into selenoproteins, which, among other functions, participate in redox homeostasis and the metabolism of thyroid hormones. On the other hand, Se in higher concentrations can exert strong toxicity, in part by contributing to the formation of ROS [138]. Generally, it is shown that selenium compounds such as KSeCN can serve as antioxidants, impacting redox homeostasis and cellular health [139]. Although KSeCN is mostly known as an antioxidant and it has not been discussed until now in the literature as a potential carcinogen, we decided to check if this NIS inhibitor is able to induce oxidative damage in the thyroid. In our study, incubation of thyroid homogenates in the presence of KSeCN increased LPO in a concentration-dependent manner (statistically significant for 500 mM). Melatonin (5 mM) and IPA (5 mM) completely prevented KSeCN-induced LPO in the thyroid [134].

Sodium chlorate (NaClO_3_), a powerful oxidizing agent, is commonly used in the production of herbicides and explosives and in the bleaching process of paper pulp. Due to its properties, it can cause oxidative stress in organisms by generating ROS. It has been shown that the reduction of chlorate (ClO_3_^−^) to chloride ions (Cl^−^) results in the formation of ROS and free radicals, which can damage macromolecules [140]. In our in vitro study, NaClO_3_ (in concentrations of 0.5–10 mM) increased LPO levels in the thyroid; however, neither melatonin nor IPA prevented these oxidative damages [134].

Two NIS inhibitors applied in our study, i.e., potassium perchlorate (KClO_4_) and potassium nitrate (KNO_3_), did not change LPO in thyroid homogenates [134]. Thus, their damaging effects in the thyroid can be checked in other experimental models.

Bisphenol A is a synthetic chemical compound extensively used in the production of polycarbonate plastics and epoxy resins, found in a wide range of consumer goods such as water bottles, food containers, and the linings of metal cans [141]. It is a well-known endocrine disruptor. At the same time, potential carcinogenic effects of bisphenol A are suggested in the literature [142]. Bisphenol A has been demonstrated to increase H_2_O_2_ production in thyrocytes in both in vivo (40 mg/kg b.w. daily for 15 days) and in vitro (1 nM) settings [143], but neither melatonin nor other antioxidants were applied in this model for checking their potential protection. In our in vitro study, bisphenol A did not affect oxidative damage to membrane lipids in porcine thyroid homogenates [134]; therefore, the potential protective effects of melatonin against LPO induced by bisphenol A are still not known. However, other in vitro models should be applied to check if bisphenol A is able to induce oxidative damage to macromolecules in the thyroid gland.

The protective effects of melatonin against oxidative damage caused by potential carcinogens (not listed in the International Agency for Research on Cancer (IARC) Monographs) are summarized in Table 2 and presented in Figure 5.

**Table 2 cancers-16-01646-t002:** Protective effects of melatonin against oxidative damage caused by potential carcinogens (not listed in the International Agency for Research on Cancer (IARC) Monographs). MDA + 4-HDA, malondialdehyde+4-hydroxyalkenals; ↑, increase.

Agent/Dose	Species/Organ/Tissue/ Cellular Compartment	Effect of Agent	Dose of Melatonin, Which Reduced or Prevented the Effect of Agent	Refs.
Potassium iodate (KIO_3_)	Porcine thyroid homogenates	↑MDA + 4-HDA	5 mM	[25,84,85]
Ferrous ion (Fe^2+^)/ferrous sulfate (FeSO_4_) 40 µM or 37.5 µM–4.8 mM	Porcine thyroid homogenates	↑MDA + 4-HDA	5 mM	[20,21]
Ammonium thiocyanate (NH_4_SCN) 250–500 mM	Porcine thyroid homogenates	↑MDA + 4-HDA	5 mM	[134]
Sodium fluoride (NaF) 25–100 mM	Porcine thyroid homogenates	↑MDA + 4-HDA	5 mM	[134]
Potassium selenocyanate (KSeCN) 500 mM	Porcine thyroid homogenates	↑MDA + 4-HDA	5 mM	[134]
Sodium chlorate (NaClO_3_) 0.5–10 mM	Porcine thyroid homogenates	↑MDA + 4-HDA	No protection	[134]

## 6. Conclusions and Future Directions

This review has underscored the potent antioxidative effects of melatonin on the thyroid gland, highlighting its protective role against carcinogen-induced oxidative stress and its potential implications in thyroid health and disease. The evidence from various experimental models supports melatonin’s capacity to mitigate oxidative damage, the process being a critical factor in the pathogenesis of thyroid disorders, cancer included. However, the scarcity of human studies marks a significant gap in our understanding and application of these findings in clinical settings.

Future research should prioritize clinical trials to establish the therapeutic efficacy and safety of melatonin in human individuals with thyroid disorders. It is imperative to explore the molecular mechanisms underlying melatonin’s protective effects in greater detail, to identify potential targets for intervention. Moreover, longitudinal studies assessing the long-term outcomes of melatonin supplementation in individuals at risk of thyroid diseases could provide invaluable insights. The integration of melatonin into therapeutic regimens for thyroid conditions necessitates a multidisciplinary approach, combining insights from endocrinology, oncology, and pharmacology, to harness its full potential as a noninvasive, natural antioxidant in the prevention and treatment of thyroid diseases.

## Figures and Tables

**Figure 1 cancers-16-01646-f001:**
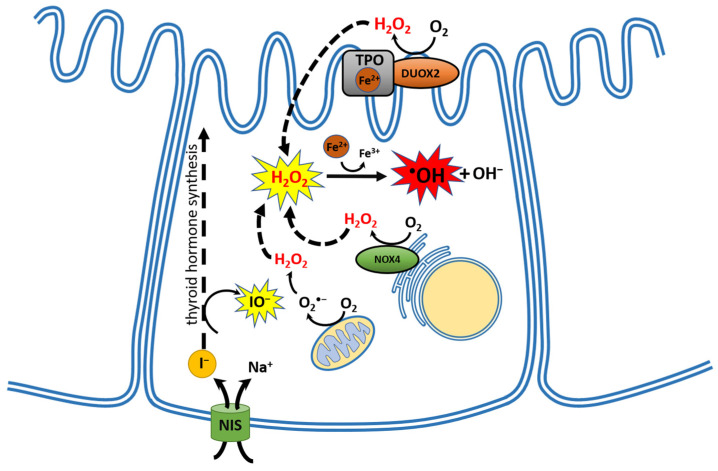
Oxidative nature of the thyroid follicular cell (thyrocyte). Only documented reactive oxygen species (ROS) or free radicals, present inside the thyroid follicular cell, are marked in the scheme. Fe^2+^, ferrous ion; Fe^3+^, ferric ion; H_2_O_2_, hydrogen peroxide; O_2_, molecular oxygen; O_2_•^−^, superoxide anion radical; •OH, hydroxyl radical; OH^−^, hydroxide ion; TPO, thyroid peroxidase; DUOX2, dual oxidase 2; NOX4, NADPH oxidase 4; IO^−^, hypoiodite anion; I^−^, iodide ion; NIS, sodium/iodide symporter; Na^+^, sodium ion.

**Figure 2 cancers-16-01646-f002:**
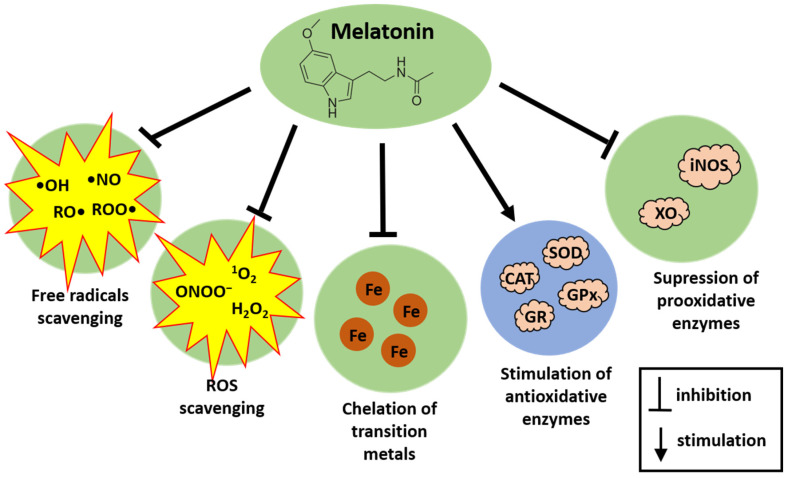
Antioxidative actions of melatonin. •OH, hydroxyl radical; RO•, alkoxy radical; ROO•, peroxy radical; •NO, nitric oxide radical; H_2_O_2_, hydrogen peroxide; ^1^O_2_, singlet oxygen; ONOO^−^, peroxynitrite anion; SOD, superoxide dismutase; GPx, glutathione peroxidase; GR, glutathione reductase; CAT, catalase; XO, xanthine oxidase; iNOS, inducible nitric oxide synthase; Fe, iron.

**Figure 3 cancers-16-01646-f003:**
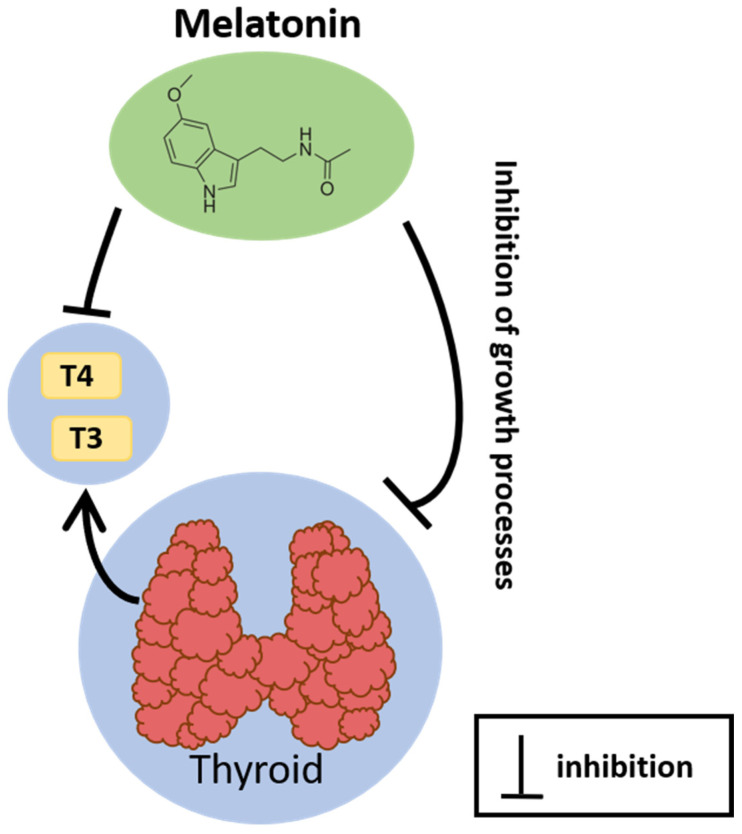
The relationship between the thyroid and melatonin documented in experimental studies. T4, thyroxine; T3, triiodothyronine.

**Figure 4 cancers-16-01646-f004:**
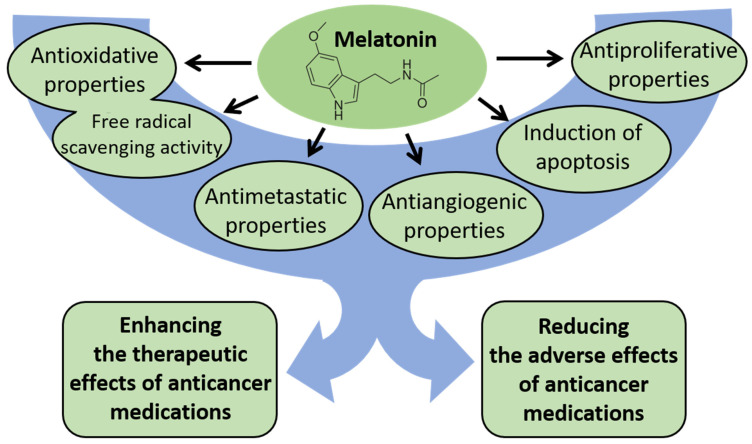
The mechanisms of the potential protective effects of melatonin against the process of carcinogenesis.

**Figure 5 cancers-16-01646-f005:**
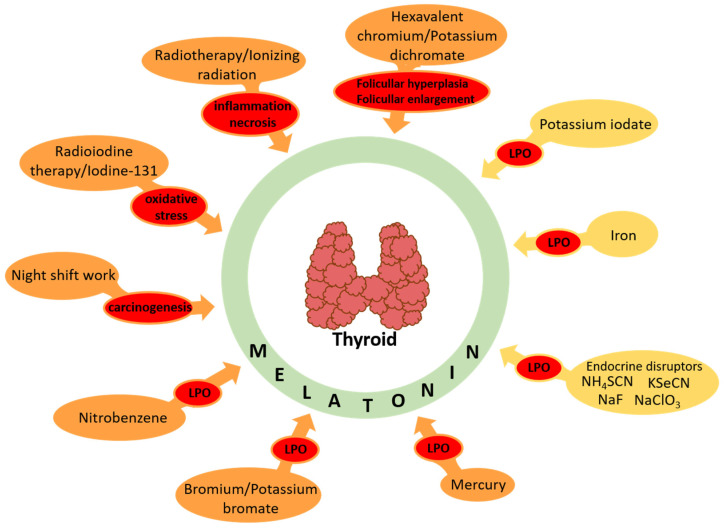
Protective effects of melatonin against documented/potential carcinogens in the thyroid. LPO, lipid peroxidation. Actions of particular documented/potential carcinogens (as well as adequate references) are presented in Table 1 and Table 2.

**Table 1 cancers-16-01646-t001:** Protective effects of melatonin against oxidative damage caused by documented carcinogens (listed in the International Agency for Research on Cancer (IARC) Monographs). MDA, malondialdehyde; MDA + 4-HDA, malondialdehyde + 4-hydroxyalkenals; SOD, superoxide dismutase; GPx, glutathione peroxidase; GR, glutathione reductase; CAT, catalase; GSH, glutathione; ip, intraperitoneally; ↑, increase; ↓, decrease.

IARC Group	Agent/Dose	Species/Organ/ Tissue/Cellular Compartment	Effect of Carcinogen	Dose of Melatonin, Which Reduced or Prevented the Effect of Carcinogen	Refs.
1	Hexavalent chromium/potassium dichromate 25 mg/kg/day for 2 months	Adult male albino rats of Wistar strain	Follicular hyperplasia, follicular enlargement	10 mg/kg/day for 2 months	[92]
1	Radiotherapy/ionizing radiation total dose of 16–18 Gy	Adult female rats	Increased inflammation, vacuolization, degradation, swelling, and necrosis in the thyroid gland	10–50 mg/kg 10–15 min before exposure	[79,80]
1	Radioiodinetherapy /iodine-131 111 MBq	Adult rats	Oxidative stress parameters in the liver	12 mg/kg/day	[101]
	Radioiodinetherapy /iodine-131 370 to 740 MBq	Hyperthyroid patients	Chromosomal damage in lymphocyte	300 mg	[102]
2A	Night-shift work	Humans	Indirect evidence on carcinogenesis	Not documented in the literature	-
2B	Nitrobenzene 7.5–10.0 mM	Porcine thyroid homogenates	↑MDA + 4-HDA in the thyroid	0.0001 mM	[112]
2B	Bromium/potassium bromate 110 mg/kg	Adult rats	↑MDA + 4-HDA in the thyroid	15 mg/kg twice daily for 10 days	[116]
3	Mercury/mercury chloride 2 and 4 mg/kg, orally	Adult male albino rats of Wistar strain	↑MDA ↓SOD, ↓CAT, ↓GPx, ↓GR, ↓GSH	5 mg/kg, ip	[125]

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
