# Peer review of "Protective Effects of Melatonin against Carcinogen-Induced Oxidative Damage in the Thyroid"

_cancers, 2024, doi:10.3390/cancers16091646_

Round 1

Reviewer 1 Report

Comments and Suggestions for Authors

This Review focuses on the protective role of melatonin against oxidative stress induced by carcinogen and its implications in thyroid health and disease.

In particular, authors provide a detailed introduction about the oxidative nature of thyroid and the antioxidant role of melatonin in cancer, in particular in thyroid cancer; in this ambit, authors describe the protective role of melatonin from potential carcinogens in the thyroid (for instance, hexavalent chromium/potassium dichromate, radiotherapy/ionizing radiation, potassium iodate, etc.).  

The Review is well organized, clearly written and well referenced. It offers a good overview about an original topic that could be extended to other human diseases.

Comments on the Quality of English Language

English requires minor editing.

Author Response

This Review focuses on the protective role of melatonin against oxidative stress induced by carcinogen and its implications in thyroid health and disease.

In particular, authors provide a detailed introduction about the oxidative nature of thyroid and the antioxidant role of melatonin in cancer, in particular in thyroid cancer; in this ambit, authors describe the protective role of melatonin from potential carcinogens in the thyroid (for instance, hexavalent chromium/potassium dichromate, radiotherapy/ionizing radiation, potassium iodate, etc.).  

The Review is well organized, clearly written and well referenced. It offers a good overview about an original topic that could be extended to other human diseases.

We would like to thank the Reviewer very much for this positive review.

Reviewer 2 Report

Comments and Suggestions for Authors

Reviewing the manuscript entitled, “Protective Effects of Melatonin Against Carcinogen-Induced Oxidative Damage in the Thyroid” by StÄ™pniak J et al., this is a review article focusing on relationship between melatonin functions and oxidative damage in the thyroid. Although this is an interesting manuscript, the authors need to respond the following concerns.

 The flow of chapters 1 to 6 is good. Easy to read. However, there is only an illustration of the physiological functions of the thyroid gland in Chapter 1. The authors mentioned about melatonin in the chapter 2 and about relationship thyroid and melatonin in the chapter 3 and describe multiple protective effects of melatonin in the chapter 4. I strongly recommend you attachment of illustrations in these chapters to further enhance the reader's understanding.

 In the chapter 4, I get the impression that the various protective effects of melatonin are somewhat narrative. The authors should provide the pharmacological background regarding these mechanisms.

 In the chapter 5, the authors described evidence on protective effects of melatonin against oxidative damage to macromolecules caused by documented/potential carcinogens in the thyroid. The research results so far are summarized in a table, but it is difficult to understand. As mentioned above, the authors need to attach illustration in the chapter 3, and using it, you should illustrate the protective effects of melatonin against carcinogens in a schematic way.

Author Response

Reviewing the manuscript entitled, “Protective Effects of Melatonin Against Carcinogen-Induced Oxidative Damage in the Thyroid” by StÄ™pniak J et al., this is a review article focusing on relationship between melatonin functions and oxidative damage in the thyroid. Although this is an interesting manuscript, the authors need to respond the following concerns.

The flow of chapters 1 to 6 is good. Easy to read. However, there is only an illustration of the physiological functions of the thyroid gland in Chapter 1. The authors mentioned about melatonin in the chapter 2 and about relationship thyroid and melatonin in the chapter 3 and describe multiple protective effects of melatonin in the chapter 4. I strongly recommend you attachment of illustrations in these chapters to further enhance the reader's understanding.

According to the Reviewer’s remark, which we found to be very helpful, we have added four schematic illustrations in chapters 2,3,4, and 5.

In the chapter 4, I get the impression that the various protective effects of melatonin are somewhat narrative. The authors should provide the pharmacological background regarding these mechanisms.

We would like to thank the Reviewer very much for this remark. We have added an appropriate text in the chapter 4 (page 7, lines 260-273).

In the chapter 5, the authors described evidence on protective effects of melatonin against oxidative damage to macromolecules caused by documented/potential carcinogens in the thyroid. The research results so far are summarized in a table, but it is difficult to understand. As mentioned above, the authors need to attach illustration in the chapter 3, and using it, you should illustrate the protective effects of melatonin against carcinogens in a schematic way.

We would like to thank the Reviewer very much for this remark. According to the Reviewer’s remark, we have added schematic illustration in the chapter 5.

Reviewer 3 Report

Comments and Suggestions for Authors

Protective Effects of Melatonin Against Carcinogen-Induced Oxidative Damage in the Thyroid. by Stępniak et al.

To the Authors:

General comments:

The authors reviewed the relationship between the thyroid gland and melatonin. Possible influences on cancer therapy in humans and the overall well-being of cancer patients are also discussed. The theme of this study is interesting, and it is well-written; however, several points should be addressed to improve the manuscript.

Specific comments:

1. The relationship between thyroid disorders and other sleep-related hormones, including orexin, should be discussed. Please highlight the unique point of melatonin in thyroid disorders compared to other hormones.

2. Do the effects of melatonin vary depending on the type of thyroid cancer, such as papillary, follicular, or medullary thyroid cancer?

3. What do the authors think is the orexin function in thyroid function disorders such as Graves' disease and Hashimoto's disease? Please discuss the relationship regarding oxidative stress.

Comments on the Quality of English Language

minor

Author Response

Specific comments:

1. The relationship between thyroid disorders and other sleep-related hormones, including orexin, should be discussed.

In response to the reviewer's remark regarding the relationship between thyroid disorders and other sleep-related hormones, including orexin, it is important to note that our manuscript primarily focuses on the protective role of melatonin against oxidative stress induced by carcinogens and its implications for thyroid health and disease. Existing literature recognizes abnormal sleep duration as a significant risk factor for various adverse health outcomes, including its effect on thyroid function. Both shorter and longer sleep durations are associated with an increased risk of subclinical thyroid dysfunction compared to optimal sleep duration. Studies also indicate that hypocretin/orexin, being a sleep related neuropeptide, may play a role in modulating thyroid regulation. Although these studies suggest a potential influence of hypocretin on thyroid function, the exact nature of this interaction – whether excitatory or inhibitory – has yet to be definitively established. This connection is further complicated in conditions such as narcolepsy, where hypocretin loss is a fundamental pathophysiological mechanism. Narcolepsy patients exhibit significantly lower levels of free thyroxine (FT4) compared to controls, highlighting a potential link between hypocretin levels and thyroid hormone regulation that could influence the severity and neuropsychological functions of narcolepsy patients. However, given that the precise mechanisms, through which sleep-related hormones like hypocretin/orexin affect thyroid function are not thoroughly explored, and that our manuscript does not focus on this aspect, we have decided not to include such information in the current revision. Nonetheless, we are willing to add the suggested data in the next revision if the reviewer still requires.

Please highlight the unique point of melatonin in thyroid disorders compared to other hormones.

It is worth stressing that melatonin stands out among various hormones implicated in thyroid health due to its potent antioxidative effects, which are critical in the context of thyroid disorders. Unlike many other hormones that influence thyroid function through direct regulatory mechanisms, melatonin primarily contributes through its ability to modulate oxidative stress, a key factor in thyroid physiology and probably thyroid pathology.

Thyroid disorders, including both hypothyroidism and hyperthyroidism, are often accompanied by an increased oxidative stress burden due to abnormal metabolic rates associated with synthesis, degradation, and peripheral effects of thyroid hormones. The fact that melatonin is a highly effective free radical scavenger and it is able to stimulate antioxidative enzymes but inhibit prooxidative enzymes, this indoleamine can be particularly beneficial in the context of thyroid diseases. It helps to protect thyroid cells from oxidative damage, which can lead to cellular dysfunction or death, and plays a role in preventing the onset and progression of thyroid diseases. Therefore, it is highly probable that melatonin affects the thyroid gland using mostly mechanisms other than endocrine mechanisms, which distinguishes this hormone from other hormones acting via binding to specific receptors. Summarizing this issue, the potent antioxidative properties of melatonin set it apart from other hormones that influence thyroid function, offering a distinctive therapeutic potential in managing thyroid disorders. 

We have added an appropriate text in the chapter 2 (page 5,  lines 196-213)

2. Do the effects of melatonin vary depending on the type of thyroid cancer, such as papillary, follicular, or medullary thyroid cancer?

We would like to thank the Reviewer very much for this question. Therefore we have added an appropriate text in Chapter 4 (page 8, lines 319-327).

“It is worth considering to what extent melatonin would be effective in particular types of thyroid cancer. The mechanisms behind the anticancer effects of melatonin are not sufficiently understood to allow for comparisons across such diverse cancers as papillary and follicular cancer, both being differentiated thyroid cancer formed from thyroid follicular cells, and medullary thyroid cancer formed from C cells. Additionally, both types of thyroid cells possess the machinery for thyroid hormone synthesis as well as melatonin receptors.  Therefore, it is currently challenging to conclusively determine if the effects of melatonin may vary specifically among these types of thyroid cancer due to the limited understanding of its underlying anticancer mechanisms”

 3. What do the authors think is the orexin function in thyroid function disorders such as Graves' disease and Hashimoto's disease? Please discuss the relationship regarding oxidative stress.

It has been shown that treatment with orexin A can significantly reduce the secretion of pro-inflammatory cytokines such as IL-1β, IL-6, and IL-8, as well as the production of reactive oxygen species (ROS) [Sun M. et al., Orexin A may suppress inflammatory response in fibroblast-like synoviocytes. Biomed Pharmacother. 2018;107:763-768. doi:10.1016/j.biopha.2018.07.159]. This suggests that orexin may modulate the inflammatory and oxidative pathways that are exacerbated in thyroid disorders. The potential of orexin to mitigate these pathways offers a promising avenue for therapeutic research, targeting the reduction of oxidative stress and inflammation, which are pivotal in the progression of both Graves' disease and Hashimoto's disease. However, to our knowledge, there is no experimental evidence directly linking these effects to thyroid disorders. As the above considerations are beyond the scope of our review, we have decided not to include them in the current revision. However, we are willing to add the suggested information in the next revision if the reviewer still requires.
